# ChatGPT-4o and 4o1 Preview as Dietary Support Tools in a Real-World Medicated Obesity Program: A Prospective Comparative Analysis

**DOI:** 10.3390/healthcare13060647

**Published:** 2025-03-16

**Authors:** Louis Talay, Leif Lagesen, Adela Yip, Matt Vickers, Neera Ahuja

**Affiliations:** 1Faculty of Arts and Social Sciences, University of Sydney, Sydney, NSW 2050, Australia; 2Dietitians Australia, Phillip, ACT 2606, Australia; leif.apd@gmail.com; 3Harvard T.H. Chan School of Public Health, Harvard University, Boston, MA 02115, USA; 4Eucalyptus, Sydney, NSW 2000, Australia; 5Hospital Medicine Division, Department of Medicine, Stanford University, Stanford, CA 94305, USA; nkahuja@stanford.edu

**Keywords:** ChatGPT-4o, dietetics, health coaching, digital weight loss services, weight loss medications

## Abstract

Background/Objectives: Clinicians are becoming increasingly interested in the use of large language models (LLMs) in obesity services. While most experts agree that LLM integration would increase access to obesity care and its efficiency, many remain skeptical of their scientific accuracy and capacity to convey human empathy. Recent studies have shown that ChatGPT-3 models are capable of emulating human dietitian responses to a range of basic dietary questions. Methods: This study compared responses of two ChatGPT-4o models to those from human dietitians across 10 complex questions (5 broad; 5 narrow) derived from patient–clinician interactions within a real-world medicated digital weight loss service. Results: Investigators found that neither ChatGPT-4o nor Chat GPT-4o1 preview were statistically outperformed (*p* < 0.05) by human dietitians on any of the study’s 10 questions. The same finding was made when scores were aggregated from the ten questions across the following four individual study criteria: scientific correctness, comprehensibility, empathy/relatability, and actionability. Conclusions: These results provide preliminary evidence that advanced LLMs may be able to play a significant supporting role in medicated obesity services. Research in other obesity contexts is needed before any stronger conclusions are made about LLM lifestyle coaching and whether such initiatives increase care access.

## 1. Introduction

### 1.1. Study Context

Obesity has arguably become the most serious global health problem [1]. Over forty percent of the world’s adults were overweight in 2022, and sixteen percent were living with obesity [1]. These figures tend to be even higher in Anglo-Saxon countries such as Australia, where adult overweight and obesity rates reached sixty-six and thirty-two percent in 2022, respectively [2]. Large analyses have revealed that women, people residing in households with upper-middle incomes, and people in the Americas, Polynesia, or Micronesia are overrepresented in obesity statistics [3,4]. As a chronic complex illness, obesity requires continuous multidisciplinary treatment [5]. Patients with significant work and/or family commitments have historically struggled to adhere to such treatment in face-to-face (F2F) settings, where they often have to travel between clinics and compete for limited feasible appointment times [6,7]. Digital weight-loss services (DWLSs) have recently emerged as a promising solution to these access barriers to obesity care [8,9,10]. Certain DWLSs offer asynchronous options, which allow patients to access care at a time and place convenient to them [10]. However, a drawback of these asynchronous care models is that patients have to wait for a member of their multidisciplinary team (MDT) to respond to their input. Against the backdrop of this issue, stakeholders have become interested in the potential of large language models (LLMs) to deliver lifestyle coaching in DWLSs [11,12]. Lifestyle coaches (typically dietitians or nutritionists) should have the most contact with patients out of all MDT members in continuous obesity programs. If LLMs could deliver high-quality lifestyle coaching support within a DWLS, patients would not only be able to access care at a time and place of their convenience, but they would also receive real-time responses to their consultation input [12]. Moreover, the use of LLMs would likely reduce the cost of DWLSs, and it would make them more accessible to lower socioeconomic groups [13]. However, as weight loss is emotive and scientific, soliciting obesity treatment advice from non-human agents carries a considerable risk to patient safety. This risk is magnified if the DWLS prescribes modern weight loss medications, such as glucagon-like peptide-1 receptor agonists (GLP-1 RAs), which consistently yield side effects and are understudied in real-world settings [10,14].

### 1.2. Related Studies

ChatGPT is an LLM that uses artificial intelligence to generate human-like responses to a user’s text-based prompts [15]. The model has been studied across a range of healthcare functions, including language translation, clinical diagnoses, and scientific writing [16,17,18]. Recent research has also explored the safety and utility of ChatGPT dietary coaching [19]. In addition to the cost and efficiency benefits discussed above, scholars have argued that ChatGPT can be an excellent supporting tool for dietitians who need to obtain quick secondary opinions or generate fast summaries for patients seeking additional information [11]. A 2025 systematic review of ChatGPT’s reliability in providing dietary recommendations found that fifteen studies had hitherto focused on the LLM’s performance in isolation, four had provided descriptive insights, and five had compared the LLM with human dietitians [20]. Two of the comparative studies evaluated the accuracy with which ChatGPT models could adhere to established dietary guidelines, including the US Department of Agriculture’s dietary reference intake [21] and the Mayo Clinic Renal Diet Handbook for chronic kidney disease patients [22]. The former found that the LLM had difficulty catering to vegans [21], and the latter concluded that while ChatGPT-4 had outperformed its predecessor, it still made errors identifying potassium and phosphorous content in various foods [22]. Another study found that ChatGPT-4 accurately calculated protein content in 60.4% of food items selected from a United Nations report [23]. The remaining two studies from the systematic review [20] compared ChatGPT and human responses to a list of obesity patient questions [13,19]. The first of these qualitatively analyzed responses to ten distinct prompts and concluded that the frequency with which ChatGPT-3.5 provided misleading information outweighed its access benefits [13]. The latter, a 2023 study by Kirk et al., quantitatively compared ChatGPT-3.5 and human dietitian responses to eight questions using a three-item metric (scientific correctness, comprehensibility, and actionability). The study reported that the LLM outperformed human dietitians across five of the study’s eight questions and achieved comparable scores for the remaining three questions [19].

However, we contend that the questions in the Kirk et al. study were too simplistic to reflect real-world patient–clinician interactions; only one of the study’s eight questions contained more than ten words and a single sentence [19]. Additionally, the assessment rubric lacked an empathy component—a feature of LLMs that multiple experts consider a key limitation relative to human clinicians [24,25,26]. Empathy is defined as the ability to understand a person’s emotions or to ‘see the world through someone else’s eyes’ [27]. Several experts stress that the term encompasses the capacity to respond to another being’s emotions in a nuanced and personable manner [24,27,28]. A 2024 systematic review of LLM capacity for empathy found that assessment measures in healthcare contexts varied considerably, ranging from qualitative analyses to the 10-item Jefferson Empathy Scale tool [27]. The investigators (Sorin et al.) encouraged future researchers to blind reviewers to response-givers (LLM vs. human) and to limit biases from lengthy responses. The Materials and Methods Section will detail how this study implemented Sorin et al.’s recommendations.

Although other recent studies have also made positive discoveries from LLM dietary coaching [29,30], negative conclusions tend to be more salient. In addition to empathy and relatability limitations, a tendency to provide inappropriate recommendations to specific or complex queries has been observed [12,31,32]. Studies have also noted that previous ChatGPT models have provided inconsistent responses to the same prompt over time, creating potential confusion, and that some of the sources they referenced were fake [31]. In an assessment of macro- and micro-nutrient advice, investigators highlighted the frequency with which ChatGPT misinterpreted decimal numbers, which led to consistent errors [13]. However, to the knowledge of the authors, all the above limitations pertain to ChatGPT-3.5 and earlier versions, rather than the latest model, ChatGPT-4o and a Beta version of its upgrade, ChatGPT-4o1 preview. Finally, the literature does not appear to have hitherto tested the competency of LLMs in providing lifestyle coaching in DWLSs that use GLP-1 RA medications. Given the large uptake of these services and GLP-1 RA medications for weight loss in general [33,34], coupled with ongoing skepticism of DWLS prescribing safety [34,35], assessing LLM coaching in this context could have significant public health implications.

### 1.3. Study Aims

This study aims to compare ChatGPT versions 4o and 4o1 preview with human dietitian responses to a set of questions from medicated DWLS patients. It is believed that the study findings will enrich the emerging literature on LLM dietetics by providing an indication of the extent to which the most advanced Chat GPT model to date can emulate dietitian responses to questions from real-world DWLS patients.

## 2. Materials and Methods

### 2.1. Study Design

Study questions were developed by a team of three dietitians from a large multinational DWLS, Eucalyptus [8]. All dietitians were accredited at Dietitians Australia and have at least two years’ experience in DWLS dietary coaching. Investigators requested that the team develop a set of ten questions based on common themes from their experiences with managing Eucalyptus DWLS patients. Investigators also requested that the team divide these questions into five broad and five narrow subject frames, using standard patient register. Questions were discussed and refined over three videoconferencing sessions. Once the set of questions was finalized, investigators reproduced them on two separate sheets as follows: one in preparation for the two ChatGPT models and the other for human dietitians. The two models included ChatGPT-4o and ChatGPT-4o1 preview.

Investigators entered a detailed prompt into ChatGPT to describe the LLM’s role as a dietitian, the characteristics of a mock patient, and the response guidelines (Appendix A). No example response was provided in the prompt, rendering it a ‘zero-shot’ prompt. Although researchers have found that one- and multi-shot prompts (prompts that include single and multiple relevant examples) generate better ChatGPT responses, the investigators felt that such prompts would not reflect initial LLM-supported DWLSs in the real world. Moreover, as this is the first study on ChatGPT version 4o in the field of dietetics, investigators considered it necessary to test the potential of the model in its standard form before adding extra levels of sophistication. Existing studies support the use of zero-shot prompting as an initial evaluation method for LLMs in healthcare [36,37].

The study questions were finalized on 9 November 2024 (Appendix B) and sent to two Eucalyptus dietitians for testing. No further changes were made, and the final set of questions were entered into the two LLMs and sent to the two human dietitians via email on 12 November 2024. Both human dietitians came from the Eucalyptus DWLS; however, neither received any information about the study prior to receiving the questions. At the time of the study, both dietitians were accredited at Dietitians Australia and had over three years’ experience in dietetics, including over 18 months experience in DWLS dietary coaching. Both received the same mock patient profile and response guidelines as the ChatGPT comparator, which included an instruction to limit responses to 200 words or less. They did not receive the detailed description of the dietitian role as this was expected as part of their employment.

Responses to all ten questions from ChatGPT and the two dietitians were forwarded to four independent dietitians for assessment. All four assessors were accredited at Dietitians Australia. Assessor 1 had over 5 years’ experience as a dietitian; assessors 2 and 3 had over 2 years’ experience; and assessor 4 was a recent graduate. The assessors were required to give a score from 0 to 10 on four criteria, with 0 representing the worst response imaginable and 10 reflecting a perfect response. The four assessment criteria were derived from the rubric used in the 2023 Kirk et al. study [19], which included scientific correctness, comprehensibility, and actionability. However, as noted in the introduction, we believed that this rubric lacked an empathy criterion, given that this response aspect has been regularly highlighted as a shortcoming of ChatGPT communication in healthcare [10,26]. Accordingly, the investigators added this criterion to the assessment rubric and made some minor modifications to the other criteria descriptions (Appendix C). We framed the empathy criterion as ‘empathy/relatability’, as we believed the latter word was necessary to emphasize the ‘nuanced and personable’ communication style discussed in the introduction. Assessors were also invited to discuss any potential strengths, weaknesses, or areas for improvement in a free-text section adjacent to each response. All assessors were blinded to the response-givers. In addition to this blinding, inter-rater variability was minimized through a clearly defined marking rubric (Appendix C), the encouragement of scoring justification (in the free-text section), and through the randomization of the order in which assessors received responses (e.g., ChatGPT-4o first, followed by dietitian 1, etc.).

### 2.2. Statistical Analysis

The median assessor scores were tabulated for all ten questions across the four respondents (two human dietitians and two LLMs). Levene’s and Shapiro–Wilk tests were conducted to determine the normality of the distribution and variance [38], and thus, whether a parametric analysis of variance (ANOVA) or Kruskal–Wallis test (non-parametric equivalent) would be used to compare the question- and criterion-based differences [39]. To perform criterion-based analyses, the assessor scores were aggregated for each of the four criteria and the differences between respondents were compared using ANOVA or Kruskal–Wallis tests. All analyses and visualizations were conducted on R Studio (version 2023.06.1 +524).

## 3. Results

Scores from all four assessors across all four criteria were grouped for each question, and Levene’s and Shapiro–Wilk tests were conducted [38]. The latter tests revealed that homogeneity of variance was violated, and thus, a Kruskal–Wallis analysis was determined as the appropriate test for the study’s endpoints [39].

First, data were aggregated from all criteria, and the median scores were calculated for each individual coach on each question. No median question score from any coach was below seven, and no statistical differences were observed between coach scores for any of the study’s conventional questions (Q1—*p* = 0.45, η² = −0.006; Q2—*p* = 0.67, η² = −0.024; Q3—*p* = 0.961, η² = −0.045; Q4—*p* = 0.055, *η*² = 0.077; Q5—*p* = 0.61, η² = −0.019). However, the Kruskal–Wallis tests found statistically significant differences in dietitian responses to questions six, seven, eight, and ten (Table 1). The eta squared (*η*²) value in questions six (0.097), seven (0.121), and ten (0.127) indicated a medium effect size, while the value in question eight (0.151) suggested a large effect.

Post hoc Dunn tests revealed which coach scores differed significantly across these four questions (using adjusted *p*-values from the Benjamini–Hochberg method to mitigate false positives). The GPT-4o model received significantly higher scores than both human coaches in question ten (Table 2) (Figure 1), and one of the human coaches in question seven (Table 3). The GPT-4o1 preview model was found to have scored significantly better than one human coach on question eight (Table 4). On question six, the significant difference stemmed from a disparity between the two human coaches (Table 5).

Response scores were also concatenated into the following four individual rubric criteria: comprehensibility, empathy/relatability, scientific correctness, and actionability. Levene’s tests revealed that homogeneity of variance was violated across all four criteria, and subsequently, Kruskal–Wallis tests were used to assess the relationship between marker-specific scores and coaches (Table 6). These tests only detected a statistically significant association in the actionability category, (X^2^[3, N = 40 = 12.726, *p* < 0.01]), whose *η*^2^ value (0.27) indicated a large effect size. A pairwise post hoc Dunn test showed that this association stemmed from the significantly higher scores received by the GPT 4o model compared to the two human coaches (Table 7) (Figure 2).

## 4. Discussion

Rising global obesity rates and ongoing advances in artificial intelligence have engendered increasing scholarly interest in the use of LLMs in dietetics and obesity services. Prior to this study, some research had shown that LLMs could provide sound dietary counseling across a range of basic questions [19]. Moreover, it had been argued that LLM deployment in obesity services had the potential to increase care access by way of enhancing scalability and reducing consumer costs, and by enabling immediate responses to patient queries, thus minimizing care barriers of consultation scheduling and wait times [11,12,13,29]. To the knowledge of the authors, this study was the first to assess the capabilities of a ChatGPT-4o model in an obesity service context. It also appears to represent the first study to assess this LLM’s capabilities in assisting patients from a medicated DWLS—a care model that is becoming increasingly popular in the current healthcare services landscape. 

The analysis found that the study’s human dietitians did not achieve a statistically higher score than either ChatGPT-4o model on any of the study’s ten questions or the four assessment criteria. The same outcome was observed in the 2023 Kirk et al. study [19], which used the Chat GPT-3.5 model; however, in that study, all eight questions used simple syntax (seven of the eight questions consisted of a single sentence, using ten words or less). By contrast, all ten questions in this study were based on patient communication in a large medicated DWLS, and they ranged from forty to ninety-two words in length. Moreover, this assessment contained an even mix of broad and narrow questions, some of which involved GLP-1 RA medications and thus an additional layer of complexity. The fact that neither Chat GPT-4o nor Chat GPT-4o1 preview was outscored by either human coach on any question suggests that these LLMs (and more advanced versions) have the potential to play a significant supporting role in medicated DWLSs. 

This assertion is arguably reinforced by the finding that neither LLM was outscored by a human coach on the aggregated empathy/relatability criterion. While there has been general consensus around the utility of LLMs within a narrow healthcare scope, many experts have expressed concerns about their capability to convey empathy in patient consultations [18,24,27], which various studies have supported [28,29]. This study indicates that LLMs can in fact exhibit this capability across a range of complex queries. Moreover, the discovery that the standard Chat GPT-4o model achieved statistically higher scores for the aggregated actionability criterion may reflect greater reliability in this area of lifestyle coaching, but this needs to be further investigated before any conclusions are drawn. A possible explanation for this finding is that LLMs follow instructions without exception [15,17], whereas humans are prone to intuition and the omission of secondary information (which in certain contexts could be the inclusion of specific actions/objectives). This study was also unable to generate a clear explanation as to why the GPT-4o model outperformed human coaches more often than the more recent GPT-4o1 preview model. A feasible reason is that the GPT-4o1 preview model, while expected to be a more refined iteration of Chat GPT-4o, may not have received enough feedback at the time of the study to correct its additional features [35].

These findings could have several public health implications. Firstly, they provide preliminary evidence that advanced LLMs have the potential to support human lifestyle coaches in medicated DWLSs and thus help relieve workforce shortages and increase service scalability. Secondly, integrating LLMs into such services would enable patients to seek immediate advice at a time and place of their preference and likely reduce the temporal barrier to obesity care that many patients face [10,40]. Thirdly, the use of LLMs in DWLSs would feasibly lower their costs and enable more patients from lower socio-economic groups to access them, as these patients tend to be overrepresented in overweight and obesity statistics in Western countries [41,42]. Finally, these findings suggest that advanced LLMs could be utilized more broadly across health systems to improve efficiency. It must be stressed, however, that this study does not provide evidence that Chat GPT-4o models can replace dietitians in real-world DWLSs. Ethical considerations concerning patient privacy, potential LLM algorithmic bias, and general safety necessitate continued human oversight in all real-world weight loss interventions. As Aydin et al. (2025) assert, LLMs offer multiple benefits to healthcare interventions, but clear regulatory boundaries are needed to ensure they “serve as supportive rather than standalone resources” [43].

While the study generated important knowledge, it had multiple limitations. Firstly, LLM responses were only compared with responses from two human dietitians. Although both dietitians had had over eighteen months lifestyle coaching experience at medicated DWLSs, their responses may not be representative of coaches across this service spectrum. Secondly, questions were developed by a team of dietitians from the Eucalyptus DWLS and therefore may have missed content that features more commonly in other medicated DWLSs. Thirdly, as is the case in any subjective scoring system, this study’s scoring metric may have engendered inter-rater variability (e.g., one assessor’s interpretation of a 6, may have been considered a 7 by another). Investigators opted for a continuous 0–10 metric to align with the Kirk et al. study [19] and enable more meaningful statistical analyses. Fourthly, the study did not solicit specific feedback on empathy, and thus, the assessor scores were not enriched by qualitative assessments. And finally, as this was the first assessment of Chat GPT-4o in an obesity service context, investigators wanted to test the model in its most rudimentary state—through ‘zero-shot’ prompting. Consequently, responses did not reflect the back-and-forth communication style that would typically be observed in patient–clinician interactions of a real-world DWLS. Future investigations should seek to build upon this study’s findings by investigating the competency of LLM lifestyle coaches to engage in back-and-forth conversation after receiving multi-shot prompting. Real-world DWLSs that have already integrated AI may consider training LLMs with exemplary responses to weight-loss patient queries prior to testing. Researchers should also consider comparable analyses of LLM responses to questions developed from other medicated DWLSs.

## 5. Conclusions

Recent research has demonstrated the potential of Chat GPT-3 models as dietary support tools across a series of basic prompts. This study aimed to compare the responses of the more advanced Chat GPT-4o models to human dietitian responses across a range of complex questions based on patient–clinician interactions at a real-world medicated DWLS. The analysis found that neither Chat GPT-4o nor Chat GPT-4o1 preview were statistically outperformed by human dietitians on any of the study’s ten questions. It also generated the same findings when scores were aggregated from the ten questions across the following four individual study criteria: scientific correctness, comprehensibility, empathy/relatability, and actionability. These findings provide preliminary evidence that advanced LLMs may be able to play a significant supporting role in medicated obesity services. They should not, however, be interpreted as evidence that such LLMs can safely replace dietitians or clinical decision makers in DWLSs, as the study did not simulate the back-and-forth dialogue and nuanced decision making that characterize real-world patient–clinician interactions. To better approximate these interactions, future studies should incorporate multi-shot LLM prompting, fine-tuned models trained on patient-dietitian exchanges, and reinforcement learning techniques that adapt to complex consultations. Such studies should compare multiple modern LLMs to generate insights regarding potential differences. Future researchers should also seek to examine the effect of LLM integration on DWLS access. Real-world DWLSs should only use LLMs as a supportive tool until clear regulatory mechanisms are established.

## Figures and Tables

**Figure 1 healthcare-13-00647-f001:**
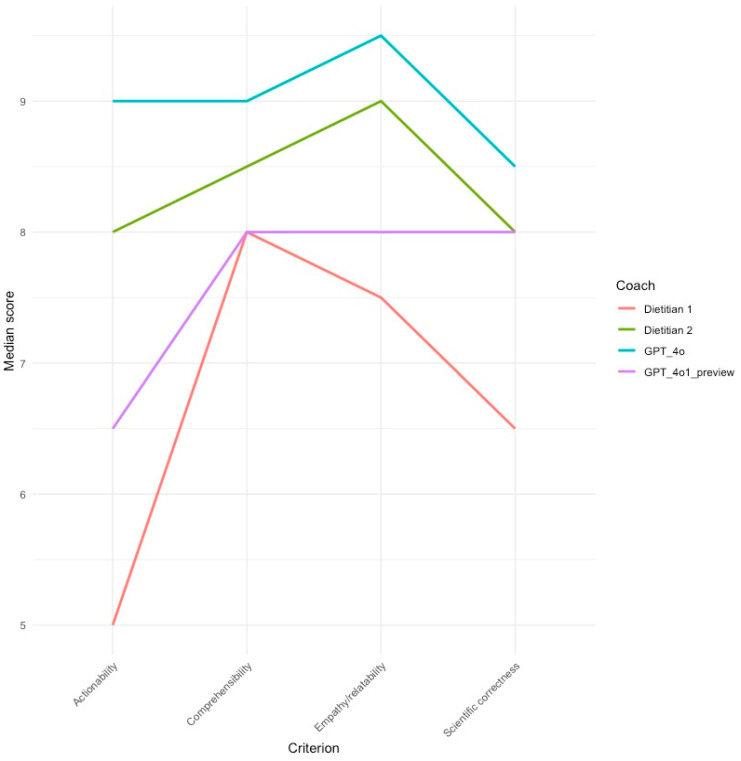
Question 10 scores by coach and criteria.

**Figure 2 healthcare-13-00647-f002:**
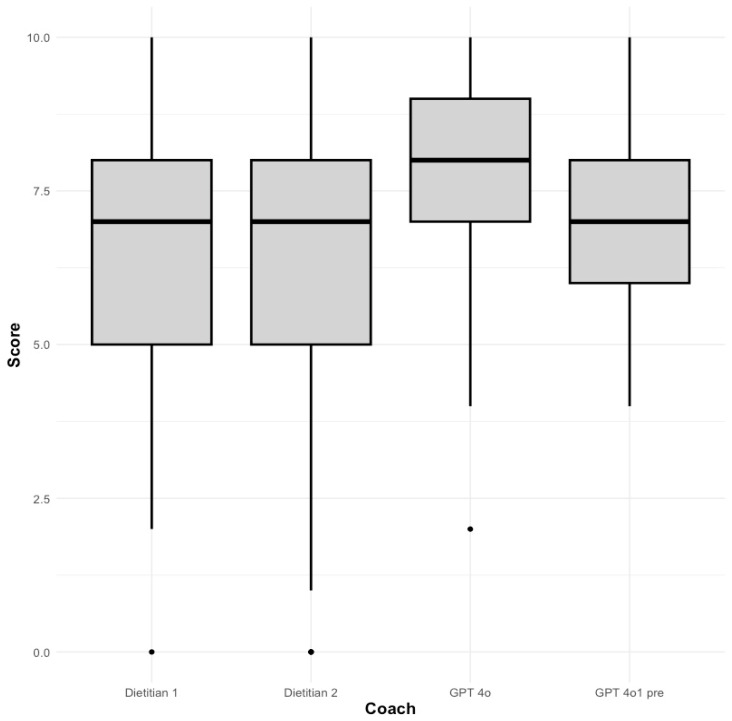
Box plot of actionability scores by coach.

**Table 1 healthcare-13-00647-t001:** Median scores and correlation statistics for each question.

	Human Dietitian 1	Human Dietitian 2	GPT-4o	GPT-4o1 Preview	Eta Squared (η²)	*p*-Value
Question 1	8 (CI 6.52, 8.7)	7 (CI 6.19, 7.72)	8 (CI 6.83, 8.97)	8 (CI 6.34, 8.65)	−0.006	0.449
Question 2	8.5 (CI 8.13,9.23)	8 (CI 7.24, 8.35)	7 (CI 6.14, 8.12)	8 (CI 7.42, 8.8)	−0.024	0.671
Question 3	7.5 (CI 6.42, 8.63)	7 (CI 6.54, 7.6)	7 (CI 6.23, 7.92)	8 (CI 6.94, 8.42)	−0.045	0.961
Question 4	7 (CI 6.22, 7.73)	5 (CI 4.60, 5.37)	7.5 (CI 6.55, 8.34)	8 (CI 6.71, 8.21)	0.077	0.055
Question 5	8 (CI 7.1, 9.03)	7.5 (CI 6.85, 8.3)	9 (CI 7.88, 9.45)	8.5 (CI 7.46, 9.04)	−0.019	0.608
Question 6	7 (CI 5.96, 8.14)	9 (CI 8.66, 9.41)	8 (CI 7.28, 9.11)	7 (CI 6.64, 8.49)	0.097	0.032 *
Question 7	7 (CI 6.35, 7.88)	8 (CI 7.04, 8.79)	9 (CI 8.14, 9.73)	7.5 (CI 5.32, 9.12)	0.121	0.017 *
Question 8	8 (CI 7.28, 9.04)	7.5 (CI 6.22, 8.65)	8 (CI 7.04, 9.19)	9 (CI 7.52, 9.41)	0.151	0.007 **
Question 9	9 (CI 8.43, 9.55)	8 (CI 7.2, 8.65)	10 (CI 10,10)	9 (CI 7.93, 9.84)	0.012	0.294
Question 10	7 (CI 6.3, 8.09)	8 (CI 7.08, 8.87)	9 (CI 8.7, 9.42)	7.5 (CI 6.87, 8.38)	0.127	0.014 *

Note: CI = 95% confidence interval; * *p* < 0.05, ** *p* < 0.01.

**Table 2 healthcare-13-00647-t002:** Post hoc Dunn test results—question 10 response score by coach.

Response Score	Coach	Levels	Z Score	*p-Adjusted* Value
		GPT-4o—GPT-4o1	0.97	0.396
		GPT-4o1—Human 1	−1.88	0.121
		GPT-4o—Human 1	−2.85	0.026 *
		GPT-4o1—Human 2	−1.52	0.191
		GPT-4o—Human 2	−2.50	0.038 *
		Human 1—Human 2	0.35	0.72

Note: * *p* < 0.05.

**Table 3 healthcare-13-00647-t003:** Post hoc Dunn test results—question 7 response score by coach.

Response Score	Coach	Levels	Z Score	*p-Adjusted* Value
		GPT-4o—GPT-4o1	1.98	0.144
		GPT-4o—Human 1	3.12	0.011 *
		GPT-4o1—Human 1	1.14	0.305
		GPT-4o—Human 2	1.25	0.314
		GPT-4o1—Human 2	−0.72	0.469
		Human 1—Human 2	−1.86	0.125

Note: * *p* < 0.05.

**Table 4 healthcare-13-00647-t004:** Post hoc Dunn test results—question 8 response score by coach.

Response Score	Coach	Levels	Z Score	*p-Adjusted* Value
		GPT-4o—GPT-4o1	2.12	0.102
		GPT-4o—Human 1	−1.32	0.225
		GPT-4o1—Human 1	−3.43	0.004 **
		GPT-4o—Human 2	0.14	0.891
		GPT-4o1—Human 2	−1.98	0.095
		Human 1—Human 2	1.45	0.219

Note: ** *p* < 0.01.

**Table 5 healthcare-13-00647-t005:** Post hoc Dunn test results—question 6 response score by coach.

Response Score	Coach	Levels	Z Score	*p-Adjusted* Value
		GPT-4o—GPT-4o1	0.35	0.724
		GPT-4o—Human 1	1.11	0.4
		GPT-4o1—Human 1	0.76	0.538
		GPT-4o—Human 2	1.75	0.159
		GPT-4o1—Human 2	2.11	0.106
		Human 1—Human 2	−2.86	0.025 *

Note: * *p* < 0.05.

**Table 6 healthcare-13-00647-t006:** Kruskal–Wallis test results—difference in coach scores by criterion.

Variable	N	*η²*	*p*-Value
Comprehensibility	40	−0.05	0.74
Empathy/Relatability	40	−0.05	0.79
Scientific correctness	40	0.04	0.22
Actionability	40	0.27	0.005 **

Note: ** *p* < 0.01.

**Table 7 healthcare-13-00647-t007:** Post hoc Dunn test results—actionability score by coach.

Response Score	Coach	Levels	Z Score	*p-Adjusted* Value
		GPT-4o—GPT-4o1	1.78	0.149
		GPT-4o—Human 1	3.26	0.007 **
		GPT-4o1—Human 1	1.48	0.209
		GPT-4o—Human 2	2.85	0.013 *
		GPT-4o1—Human 2	1.07	0.341
		Human 1—Human 2	−0.41	0.683

Note: * *p* < 0.05, ** *p* < 0.01.

## Data Availability

The raw data supporting the conclusions of this article will be made available by the authors upon request.

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
