# Peer review of "ChatGPT-4o and 4o1 Preview as Dietary Support Tools in a Real-World Medicated Obesity Program: A Prospective Comparative Analysis"

_healthcare, 2025, doi:10.3390/healthcare13060647_

Round 1
Reviewer 1 Report
Comments and Suggestions for Authors
- The extra punctuation before "Methods" in the abstract should be removed.
- A "Related Studies" section can be added to the study, where studies related to different areas of ChatGPT, especially the diet coaching problem, are examined.
- More details about the proposed method should be given in the introduction section. At the end of this section, one sentence should definitely be given for each step of the method.
- The contribution of the method to the ChatGPT field should be emphasized.
- The results section of the study is very weak. The results are given in tables one after the other. However, none of the tables are interpreted. What do the values ​​in the tables mean? A comprehensive analysis should be done.
- The resolution of Figures 1 and 2 is very low.
- The "Conclusion" section should address the limitations of the study and explain how these limitations will guide future studies.
Author Response
Comment: - The extra punctuation before "Methods" in the abstract should be removed.
Response: Thank you for noticing this. The extra full stop has now been removed.
Comment:
- A "Related Studies" section can be added to the study, where studies related to different areas of ChatGPT, especially the diet coaching problem, are examined.
Response: Thank you for this excellent suggestion. We have now added 13 lines of text, reviewing studies relevant two different areas of LLMs. Including LLM safety (end of paragraph 1 in the intro):
“However, as weight loss is emotive and scientific, soliciting obesity treatment advice from non-human agents carries a considerable risk to patient safety. This risk is magnified if the DWLS prescribes modern weight-loss medications, such as glucagon-like peptide-1 receptor agonists (GLP-1 RAs), which yield consistent side effects and are understudied in real-world settings [10,14].”
And LLM capacity for empathy (end of paragraph 2 in the intro):
“Empathy is defined as the ability to understand a person’s emotions or to ‘see the world through someone else’s eyes’ [23]. Several experts stress that the term encompasses a capacity to respond to another being’s emotions in a nuanced and personable manner [20,23,24].A 2024 systematic review of LLM capacity for empathy found that assessment measures in healthcare contexts varied considerably, ranging from qualitative analyses to the 10-item Jefferson Empathy Scale tool [23]. The investigators (Sorin et al.) encouraged future researchers to blind reviewers to response-givers (LLM vs human) and to limit biases from lengthy responses.”
Comment:- More details about the proposed method should be given in the introduction section. At the end of this section, one sentence should definitely be given for each step of the method.
Response: Thank you for this suggestion. We have now added the following 3 sentences to the end of the 2nd paragraph of the introduction, which are relevant to the methods:
“A 2024 systematic review of LLM capacity for empathy found that assessment measures in healthcare contexts varied considerably, ranging from qualitative analyses to the 10-item Jefferson Empathy Scale tool [23]. The investigators (Sorin et al.) encouraged future researchers to blind reviewers to response-givers (LLM vs human) and to limit biases from lengthy responses. The methods section will detail how this study implemented Sorin et al’s recommendations.”
As indicated in the final sentence, we believe that the detailed explanation of study methods belongs in the methods section.
Following this we have added the following details to the methods section:
- “Both received the same mock patient profile and response guidelines as the ChatGPT comparator, which included an instruction to limit responses to 200 words or less.”
- “All assessors were blinded to response-givers. In addition to this blinding, inter-rater variability was minimized through a clearly defined marking rubric (Appendix C), the encouragement of scoring justification (in the free-text section), and through the randomization of the order in which assessors received responses (e.g., ChatGPT-4o first, followed by dietitian 1, etc).”
Comment:
- The contribution of the method to the ChatGPT field should be emphasized.
Response:
Thank you for this important recommendation. In the 4th paragraph of the discussion section, we emphasize that the study, and thus the method, “provide preliminary evidence that advanced LLMs have the potential to support human lifestyle coaches in medicated DWLSs and thus help relieve workforce shortages and increase service scalability” (which we reinforce in the conclusion).
We also add that this type of LLM integration (supportive tool) could increase patient temporal and financial access.
We have now added 3 sentences to the end of the paragraph to bolster the main point. They read as follows:
“It must be stressed, however, that this study does not provide evidence that Chat GPT-4o models can replace dietitians in real-world DWLSs. Ethical considerations concerning patient privacy, potential LLM algorithmic bias and general safety necessitate continued human oversight in all real-world weight loss interventions. As Aydin et al (2025) assert, LLMs offer multiple benefits to healthcare interventions, but clear regulatory boundaries are needed to ensure they “serve as supportive rather than standalone resources”[37].”
Comment:
- The results section of the study is very weak. The results are given in tables one after the other. However, none of the tables are interpreted. What do the values ​​in the tables mean? A comprehensive analysis should be done.
Response: Thank you for this excellent suggestion. We have now added the following content to our results section:
- Parenthesized p-values for all 5 conventional questions (paragraph 2)
- A summary sentence of effect sizes for the questions in which a sig difference was detected between coaches (end of paragraph 2): “The eta-squared (η²) value in questions six (0.097), seven (0.121) and ten (0.127) indicated a medium effect size, while the value in question eight (0.151) suggested a large effect.”
- Eta-squared values (effect sizes) for all 10 questions in table 1
- Confidence intervals across all coaches and questions in table 1
- A clause reporting the effect size of the difference observed in the Actionability category(end of paragraph 4) – “… whose η² value (0.27) indicated a large effect size.”
- Eta-squared values (effect sizes) for all 4 criteria in table 6
- Replaced the line graph in Figure 2 with a box plot to capture distribution and median instead of just the latter
Now the differences (and their strength) across the 10 questions are summarized in paragraph 2 and table 1. Paragraph 3 then reports the results of the post-hoc Dunn tests for all the questions in which a significant difference was detected (q10 – table 2 (and figure 1); q7- table 3; q8-table 4; q6 -table 5). Paragraph 4 summarizes the results of the 4 concatenated rubric criteria, with table 6 reporting all 4 criteria, table 7 showing where the differences stemmed from in the one significant criterion, and figure 2 visualizing these differences.
Comment:
- The resolution of Figures 1 and 2 is very low.
Response: Thank you for noticing this. We have now increased the resolution of figures 1 and 2.
Comment:
- The "Conclusion" section should address the limitations of the study and explain how these limitations will guide future studies.
Response: Thank you for this excellent suggestion. We have now added a sentence to give further detail on the design of future LLM dietetics studies that can address the stated limitations of our study.
“…the study did not simulate the back-and-forth dialogue and nuanced decision making that characterize real-world patient-clinician interactions. To better approximate these interactions, future studies should incorporate multi-shot LLM prompting, fine-tuned models trained on patient-dietitian exchanges, and reinforcement learning techniques that adapt to complex consultations. Such studies should compare multiple modern LLMs to generate insights on potential differences.”
Reviewer 2 Report
Comments and Suggestions for Authors
The manuscript presents a comparative analysis of ChatGPT-4o and 4o1 preview versus human dietitians in a medicated obesity program. The study is highly relevant in the context of AI integration in digital healthcare services, addressing concerns regarding accuracy, empathy, and clinical applicability. However, the manuscript requires clarifications, refinements, and structural improvements to enhance its scientific robustness.
Title and Abstract:
The current title is long and ambiguous. Consider rephrasing to highlight the study's primary contributions.
Lines 12-15: The background is too general. It should concisely explain the problem statement in obesity coaching.
Lines 16-18: Define "series of complex questions"—provide examples or categorize them.
Lines 21-24: The statistical results should include confidence intervals, effect sizes, and p-values to strengthen scientific validity.
Lines 25-27: The conclusion should clarify whether LLMs outperform or merely match human dietitians.
Introduction:
Lines 34-41: The obesity statistics are useful but should cite global, regional, and demographic variations to add depth.
Lines 42-48: Expand on how LLMs are expected to impact DWLS (Digital Weight-Loss Services). What are the potential risks of AI coaching?
Lines 49-55: Define how empathy and relatability were evaluated in prior studies and how this study improves upon past methods.
Lines 70-72: While LLM empathy concerns are noted, there is no clear explanation of how this study addresses them.
Methods:
Lines 97-102: The study design should specify whether the dietitians and LLMs had the same response constraints (word count, format).
Lines 103-115: Provide a justification for using zero-shot prompting. Previous research suggests few-shot prompting improves LLM performance.
Lines 118-122: Clarify how inter-rater reliability was maintained in scoring (e.g., did assessors undergo calibration training?).
Lines 135-138: The empathy/relatability metric is subjective—explain how assessors ensured consistency in scoring.
Lines 150-156: The statistical approach should justify the use of Kruskal-Wallis over parametric methods.
Expand on study design choices, justify statistical methods, and ensure scoring reliability is explained.
Results:
Lines 167-169: The statement "No statistical differences were observed for conventional questions" should include exact p-values and effect sizes.
Table 1 (Lines 170-171): Missing a column for confidence intervals—crucial for evaluating statistical robustness.
Lines 173-177: The Dunn post hoc analysis should specify which criteria showed significant differences.
Lines 188-193: The finding that GPT-4o scored higher in actionability requires explanation on its practical implications.
Lines 194-199: Missing a graphical visualization of model performance—suggest adding a box plot for score distributions.
Provide more granular statistical data, include confidence intervals, and add visual representation of key results.
Discussion:
Lines 214-217: Compare results with prior research to determine whether GPT-4o outperforms, matches, or underperforms human dietitians.
Lines 228-232: The claim that GPT-4o performs well in empathy/relatability lacks direct assessor feedback—consider quoting assessors.
Lines 245-254: Discuss real-world implications—would GPT-4o replace or assist dietitians? Ethical considerations?
Lines 265-269: The limitation on zero-shot prompting should discuss how real-world AI implementations would likely use prompt fine-tuning.
Strengthen the comparison with past studies, include expert perspectives, and address ethical concerns.
Conclusion:
Lines 280-283: The phrase "provides preliminary evidence" should quantify the evidence strength using statistical values.
Lines 284-287: Clarify whether GPT-4o can be used in clinical decision-making or only as an adjunct to dietitians.
Lines 288-291: The call for future research should mention what type of datasets and models should be tested.
Make the conclusion more conclusive, clarify real-world applicability, and define future research directions.
Additional Requirements:
Effect sizes and confidence intervals should be reported in all tables.
The normality assumption of Kruskal-Wallis should be clearly justified.
Ensure that Dunn test p-values are adjusted for multiple comparisons.
Abstract needs clear statistical summary.
Results should integrate figures/tables for clarity.
Discussion should emphasize clinical impact and AI ethics.
The potential biases in AI responses should be explored.
Clearly state whether GPT-4o was tested for misinformation detection.
Comments on the Quality of English Language
In addition to the structural and technical revisions, the manuscript requires several linguistic and grammatical refinements to improve clarity, coherence, and scientific professionalism.
Author Response
The manuscript presents a comparative analysis of ChatGPT-4o and 4o1 preview versus human dietitians in a medicated obesity program. The study is highly relevant in the context of AI integration in digital healthcare services, addressing concerns regarding accuracy, empathy, and clinical applicability. However, the manuscript requires clarifications, refinements, and structural improvements to enhance its scientific robustness.
Title and Abstract:
Comment: The current title is long and ambiguous. Consider rephrasing to highlight the study's primary contributions.
Response: Thank you for this excellent suggestion. We have now changed the title to the following:
“ChatGPT-4o and 4o1 preview as a dietary support tool in a real-world medicated obesity program: a prospective comparative analysis”
Comment: Lines 12-15: The background is too general. It should concisely explain the problem statement in obesity coaching.
Response: Thank you for identifying this verbosity. We have now removed 2 long clauses from the background section of the abstract. The section now reads as follows:
“Clinicians are becoming increasingly interested in the use of large language models (LLMs) in obesity services. While most experts agree that LLM integration would increase access to obesity care and its efficiency, many remain skeptical of their their scientific accuracy and capacity to convey human empathy. Recent studies have shown that ChatGPT-3 models are capable of emulating human dietitian responses to a range of basic dietary questions.”
Comment: Lines 16-18: Define "series of complex questions"—provide examples or categorize them.
Response: Thank you for noticing the vagueness of this statement. We have now categorized the questions and revised the referred text to the following:
“…across 10 complex questions (5 broad; 5 narrow) derived from patient-clinician interactions…”
Comment: Lines 21-24: The statistical results should include confidence intervals, effect sizes, and p-values to strengthen scientific validity.
Response: Thank you for this comment. We believe that providing data on CI, effect size and p-values on all 10 questions would be unusual for an abstract, and moreover, doing so would render it too long. However, we have now specified the value we consider statistically significant ‘(p < 0.05)’ to the sentence, “Investigators found that neither ChatGPT-4o nor Chat GPT-4o1 preview were statistically outperformed (p < 0.05) by human dietitians on any of the study’s 10 questions.”
Comment: Lines 25-27: The conclusion should clarify whether LLMs outperform or merely match human dietitians.
Response: Thank you for this suggestion. We state in the results sub-section that “Investigators found that neither ChatGPT-4o nor Chat GPT-4o1 preview were statistically outperformed (p < 0.05) by human dietitians on any of the study’s 10 questions”
We interpret this in the conclusion sub-section in the following way: “These results provide preliminary evidence that advanced LLMs may be able to play a significant support role in medicated obesity services.”
Emphasizing that “Research in other obesity contexts is needed before any stronger conclusions are made about LLM lifestyle coaching”
We opted for this frame rather than using terms like ‘matching’ or ‘underperforming’ because we think it is important to use softer language in preliminary AI research to minimize the chance of industry misinterpretation and inappropriate/unethical LLM integration. We reinforce this throughout the discussion and conclusion sections of the manuscript body.
Introduction:
Comment: Lines 34-41: The obesity statistics are useful but should cite global, regional, and demographic variations to add depth.
Response: Thank you for this valuable suggestion. We have now added the following sentence to the introduction with the relevant citations to the reference list:
“Large analyses have revealed that women, people residing in homes with upper-middle incomes, and people in Americas region, Polynesia, or Micronesia are overrepresented in obesity statistics [3,4},”
Comment: Lines 42-48: Expand on how LLMs are expected to impact DWLS (Digital Weight-Loss Services). What are the potential risks of AI coaching?
Response: Thank you for this recommendation. We have now added the following to sentences to the introduction (lines 69-73) with the relevant citations to the reference list:
“However, as weight loss is emotive and scientific, soliciting obesity treatment advice from non-human agents carries a considerable risk to patient safety. This risk is magnified if the DWLS prescribes modern weight-loss medications, such as glucagon-like peptide-1 receptor agonists (GLP-1 RAs), which yield consistent side effects and are understudied in real-world settings [10,13].”
Comment: Lines 49-55: Define how empathy and relatability were evaluated in prior studies and how this study improves upon past methods.
Response: Thank you for the important request. We have now added the following five sentences to the introduction (lines 88-96) to explain how the concept was assessed in previous studies:
Empathy is defined as the ability to understand a person’s emotions or to ‘see the world through someone else’s eyes’ [23]. Several experts stress that the term encompasses a capacity to respond to another being’s emotions in a nuanced and personable manner [20,23,24].A 2024 systematic review of LLM capacity for empathy found that assessment measures in healthcare contexts varied considerably, ranging from qualitative analyses to the 10-item Jefferson Empathy Scale tool [23]. The investigators (Sorin et al.) encouraged future researchers to blind reviewers to response-givers (LLM vs human) and to limit biases from lengthy responses. The methods section will detail how this study implemented Sorin et al’s recommendations.
As indicated in the final sentence, we have added text to the Methods section to explain how this study builds upon existing LLM empathy research. This includes:
- a clause explaining the response length limitation: …” which included an instruction to limit responses to 200 words or less.”
- A sentence explaining why we framed empathy as ‘empathy/relatability’ – “We framed the empathy criterion as ‘empathy/relatability’ as we believed the latter word was necessary to emphasize the ‘nuanced and personable’ communication style discussed in the introduction.”
- And a sentence to inform readers that “All assessors were blinded to response-givers.
Comment: Lines 70-72: While LLM empathy concerns are noted, there is no clear explanation of how this study addresses them.
Response: Thank for highlighting this. Our response to the previous comment explains how we have now explained our means of addressing concerns around LLM empathy.
Methods:
Comment: Lines 97-102: The study design should specify whether the dietitians and LLMs had the same response constraints (word count, format).
Response: Thank you for noticing this oversight. We have now added a clause to the third paragraph of the methods section (lines 216-217) that specifies the word count constraint:
“Both received the same mock patient profile and response guidelines as the ChatGPT comparator, which included an instruction to limit responses to 200 words or less.
Comment: Lines 103-115: Provide a justification for using zero-shot prompting. Previous research suggests few-shot prompting improves LLM performance.
Response: Thank you for this recommendation. We have now included a sentence to the end of the second paragraph of the Methods section (lines 319-320) that supports our justification for zero-shot prompting:
“Although researchers have found that one- and multi-shot prompts (prompts that include single and multiple relevant examples) generate better ChatGPT responses, the investigators felt that such prompts would not reflect initial LLM-supported DWLSs in the real world. Moreover, as this is the first study of ChatGPT version 4o in the field of dietetics, investigators considered it necessary to test the potential of the model in its standard form before adding extra levels of sophistication. Existing research supports the use of zero-shot prompting as an initial evaluation method for LLMs in healthcare [32, 33].”
Comment: Lines 118-122: Clarify how inter-rater reliability was maintained in scoring (e.g., did assessors undergo calibration training?).
Response: Thank you for this valuable request. We have now clarified how inter-rater variability minimized through the inclusion of these two sentences at the bottom of the methods section:
“All assessors were blinded to response-givers. In addition to this blinding, inter-rater variability was minimized through a clearly defined marking rubric (Appendix C), the encouragement of scoring justification (in the free-text section), and through the randomization of the order in which assessors received responses (e.g., ChatGPT-4o first, followed by dietitian 1, etc).“
Comment: Lines 135-138: The empathy/relatability metric is subjective—explain how assessors ensured consistency in scoring.
Response: Thank you for this comment. As explained in the previous comment, we have now added several lines to clarify how general inter-rater variability was minimized. In regard to the empathy/relatability criterion specifically, we added a sentence a few lines earlier in the paragraph to explain our use of framing: “We framed the empathy criterion as ‘empathy/relatability’ as we believed the latter word was necessary to emphasize the ‘nuanced and personable’ communication style discussed in the introduction.”
The definition of the criterion was provided to assessors in the marking rubric (appendix C).
Comment: Lines 150-156: The statistical approach should justify the use of Kruskal-Wallis over parametric methods.
Expand on study design choices, justify statistical methods, and ensure scoring reliability is explained.
Response: Thank you for this comment. We have now clarified in the statistical analysis section that Kruskal-Wallis tests are a non-parametric equivalent to ANOVA and would be used if assumptions of normality were violated (via Levene’s and Shapiro Wilk tests:
“Levene’s and Shapiro Wilk tests were conducted to determine normality of distribution and variance, and thus whether a parametric analysis of variance (ANOVA) or Kruskal-Wallis test (non-parametric equivalent) would be used to compare question and criterion-based differences. “
Study design and scoring reliability are now expanded on in ways described in responses to previous comments, e.g., via;
- An explanation of the LLM empathy literature in the introduction :
“Empathy is defined as the ability to understand a person’s emotions or to ‘see the world through someone else’s eyes’ [23]. Several experts stress that the term encompasses a capacity to respond to another being’s emotions in a nuanced and personable manner [20,23,24].A 2024 systematic review of LLM capacity for empathy found that assessment measures in healthcare contexts varied considerably, ranging from qualitative analyses to the 10-item Jefferson Empathy Scale tool [23]. The investigators (Sorin et al.) encouraged future researchers to blind reviewers to response-givers (LLM vs human) and to limit biases from lengthy responses. The methods section will detail how this study implemented Sorin et al’s recommendations.” - A supporting sentence on the use of zero shot prompting in the Methods section: Existing research supports the use of zero-shot prompting as an initial evaluation method for LLMs in healthcare [32, 33].
- And a clarification of the response word limit “which included an instruction to limit responses to 200 words or less.”
The final two sentences added to the methods section explain the methods to enhance inter-rater reliability:
“All assessors were blinded to response-givers. In addition to this blinding, inter-rater variability was minimized through a clearly defined marking rubric (Appendix C), the encouragement of scoring justification (in the free-text section), and through the randomization of the order in which assessors received responses (e.g., ChatGPT-4o first, followed by dietitian 1, etc).”
Results:
Comment:
Lines 167-169: The statement "No statistical differences were observed for conventional questions" should include exact p-values and effect sizes.
Response: Thank you for this comment. We have now converted the Kruskall-Wallis chi-squared scores to eta- squared effect sizes in Table 1, and included the insignificant p-values and effect sizes for all 5 questions related to the above statement (in parentheses).
“Q1 – p = 0.45, η² = -0.006; Q2 – p = 0.67, η² = -0.024; Q3 – p = 0.961, η² = -0.045; Q4 p = 0.055, η² = 0.077; Q5 – p = 0.61, η² = -0.019 ).”
We have also added a sentence to the bottom of the first paragraph of the results section that reports the effect sizes of all questions that observed a statistically significant difference:
“The eta-squared (η²) value in questions six (0.097), seven (0.121) and ten (0.127) indicated a medium effect size, while the value in question eight (0.151) suggested a large effect.“
Along with a similar clause in the 3rd paragraph of the section to report the effect size of the observed difference in actionability scores:
“…whose η² value (0.27) indicated a large effect size.”
Comment:
Table 1 (Lines 170-171): Missing a column for confidence intervals—crucial for evaluating statistical robustness.
Response: Thank you for this excellent recommendation. Instead of adding a column for CI next to each question in the table, which would aggregate data from all coaches and assessors (and therefore fail to demonstrate where reasonable confidence, or lack thereof, exists), we have added CI values in each cell (parentheses) to capture the precision of scores on each coach for each question.
Comment:
Lines 173-177: The Dunn post hoc analysis should specify which criteria showed significant differences.
Response: Thank you for this comment. The third paragraph of the results section specifies which coach scores showed significant differences, along with the tables that pertain to each stated difference.
“Post-hoc Dunn tests revealed which coach scores differed significantly in these four questions (using adjusted p-values from the Benjamini-Hochberg method to mitigate false positives). The GPT-4o model received significantly higher scores than both human coaches in question ten (Table 2) (Figure 1), and one of the human coaches in question seven (Table 3). The GPT 4o1 preview model was found to have scored significantly better than one human coach on question eight (Table 4). On question six, the significant difference stemmed from a disparity between the two human coaches (Table 5).”
The fourth paragraph does the same for the four concatenated rubric criteria:
“These tests only detected a statistically significant association in the Actionability category, (X2[3, N = 40 = 12.726, p < 0.01]). A pairwise post hoc Dunn Test showed that this association stemmed from the significantly higher scores received by the GPT 4o model compared to the two human coaches (Table 7) (Figure 2).”
Comment:
Lines 188-193: The finding that GPT-4o scored higher in actionability requires explanation on its practical implications.
Response: Thank you for this suggestion. We have now added an extra sentence to paragraph 3 of the discussion that offers a possible explanation for this finding.
“Moreover, the discovery that the standard Chat GPT-4o model achieved statistically higher scores for the aggregated actionability criterion may reflect greater reliability in this area of lifestyle coaching but needs to be further investigated before any conclusions are drawn. A possible explanation for this finding is that LLMs follow instructions without exception [15,17], whereas humans are prone to intuition and the omission of secondary information (which in certain contexts could be the inclusion of specific actions/objectives). “
Comment:
Lines 194-199: Missing a graphical visualization of model performance—suggest adding a box plot for score distributions.
Response: Thank you for this recommendation. We are very sorry, but we weren’t quite sure which aspect of model performance you were referring to. We have now replaced our line plot of median actionability scores (Figure 2) with a box plot of their distribution. Once again, sincere apologies of we misinterpreted this recommendation.
Comment:
Provide more granular statistical data, include confidence intervals, and add visual representation of key results.
Response: Thank you for this valuable suggestion. As discussed in responses to the above comments, we have now added eta-squared effect scores and confidence interval data, along with a box plot of actionability score distribution across the 4 coaches.
Discussion:
Comment: Lines 214-217: Compare results with prior research to determine whether GPT-4o outperforms, matches, or underperforms human dietitians.
Response: Thank you for this suggestion. We have compared results with all prior research on LLM-use in dietetics throughout paragraphs 1 and 2 of the discussion, drawing a specific comparison with the only other study to have assessed LLM dietary advice across similar markers (Kirk et al’s study on GPT 3.5).
“Prior to this study, some research had shown that LLMs could provide sound dietary counselling across a range of basic questions [16]. Moreover, it had been argued that LLM deployment in obesity services had the potential to increase care access by way of enhancing scalability and reducing consumer costs, and by enabling immediate responses to patient queries, thus minimizing care barriers of consult scheduling and wait times [9,10,11,21]. To the knowledge of the authors, this study was the first to assess the capabilities of a ChatGPT-4o model in an obesity service context. It also appears to represent the first study to assess this LLM’s capabilities in assisting patients from a medicated-DWLS – a care model that is becoming increasingly popular in the current healthcare services landscape.
The analysis found that the study’s human dietitians did not achieve a statistically higher score than either ChatGPT-4o model on any of the study’s ten questions or the four assessment criteria. The same outcome was observed in the 2023 Kirk et al study [16], which used the Chat GPT-3.5 model; however, in that study, all eight questions used simple syntax (seven of the eight questions consisted of a single sentence, using ten words or less). By contrast, all ten questions in this study were based on patient communication in a large medicated DWLS, and ranged from forty to ninety-two words in length. Moreover, this assessment contained an even mix of broad and narrow questions, some of which involved GLP-1 RA medications and thus an additional layer of complexity.”
We state at the top of the second paragraph that “…dietitians “did not achieve a statistically higher score than either ChatGPT-4o model on any of the study’s ten questions or the four assessment criteria.”
We opted for this frame rather than using terms like ‘matching’ or ‘underperforming’ because we think it is important to use softer language in preliminary AI research to minimize the chance of misinterpretation and inappropriate/unethical LLM integration. We reinforce this through the final sentence of the second paragraph:
“The fact that neither Chat GPT-4o and nor Chat GPT 4-o1 preview was outscored by either human coach on any question suggests that these LLMs (and more advanced versions) have the potential to play a significant supporting role in medicated DWLSs.”
Comment: Lines 228-232: The claim that GPT-4o performs well in empathy/relatability lacks direct assessor feedback—consider quoting assessors.
Response: Thank you for highlighting this shortcoming. We have now added this to the final paragraph of the section as a limitation:
“Fourthly, the study did not solicit specific feedback on empathy and thus assessor scores were not enriched by qualitative assessments.”
Comment:
Lines 245-254: Discuss real-world implications—would GPT-4o replace or assist dietitians? Ethical considerations?
Response: Thank you for identifying this important point. We have now added 3 sentences to the end of the penultimate paragraph of the discussion section to address RW implications and ethical considerations.
“It must be stressed, however, that this study does not provide evidence that Chat GPT-4o models can replace dietitians in real-world DWLSs. Ethical considerations concerning patient privacy, potential LLM algorithmic bias and general safety necessitate continued human oversight in all real-world weight loss interventions. As Aydin et al (2025) assert, LLMs offer multiple benefits to healthcare interventions, but clear regulatory boundaries are needed to ensure they “serve as supportive rather than standalone resources”[37].”
Comment:
Lines 265-269: The limitation on zero-shot prompting should discuss how real-world AI implementations would likely use prompt fine-tuning.
Response: Thank you for this excellent suggestion. We have now revised one sentence and added a new sentence to the final paragraph of the discussion section to detail possible real-world implementations:
“Future investigations should seek to build upon this study’s findings by investigating the competency of LLM lifestyle coaches to engage in back-and-forth conversation after receiving multi-shot prompting. Real-world DWLSs that have already integrated AI may consider training LLMs with exemplary responses to weight-loss patient queries prior to testing.”
Comment:
Strengthen the comparison with past studies, include expert perspectives, and address ethical concerns.
Response: Thank you for this valuable recommendation. In addition to our comparison with previous LLM dietetics research, we have now added an expert perspective and an explanation of the study’s ethical considerations to the end of the discussion section’s penultimate paragraph.
“Ethical considerations concerning patient privacy, potential LLM algorithmic bias and general safety necessitate continued human oversight in all real-world weight loss interventions. As Aydin et al (2025) assert, LLMs offer multiple benefits to healthcare interventions, but clear regulatory boundaries are needed to ensure they “serve as supportive rather than standalone resources”[37].”
Conclusion:
Comment:
Lines 280-283: The phrase "provides preliminary evidence" should quantify the evidence strength using statistical values.
Response:
Thank you for this comment. We believe it would be unusual to restate raw statistical values in the conclusion section of a study manuscript, which is designed to focus on the interpretation and implications of the results. Additionally, because we ran so many correlation tests (10 questions and 4 criteria), we believe that restating these data (eta-squared figures and p-values) would impact the readability of this section. However, if you are adamant that they should appear here, we are more than happy to include them.
Comment: Lines 284-287: Clarify whether GPT-4o can be used in clinical decision-making or only as an adjunct to dietitians.
Response: Thank you for this important point. We have added text to the sixth sentence in the conclusion to clarify our view that the results should be interpreted as evidence that advanced LLMs (such as Chat GPT-4o) have the potential to play a significant support role in medicated obesity services, but do not in any way indicate they can replace dietitians or clinical decision makers in DWLSs.
“These findings provide preliminary evidence that advanced LLMs may be able to play a significant support role in medicated obesity services. They should not, however, be interpreted as evidence that such LLMs can safely replace dietitians or clinical decision makers in DWLSs, as the study did not simulate the back-and-forth dialogue and nuanced decision making that characterize real-world patient-clinician interactions.”
Comment:
Lines 288-291: The call for future research should mention what type of datasets and models should be tested.
Response: Thank you for this excellent suggestion. We have now added the following sentence to the end of the conclusion:
“Such studies should compare multiple modern LLMs to generate insights on potential differences.”
Comment: Make the conclusion more conclusive, clarify real-world applicability, and define future research directions.
Response: Thank you for this valuable comment. In addition to stressing the conservation interpretation of the study results (sentence 5 and 6) we have added the following sentence to the end of the conclusion:
“Real-world DWLSs should only use LLMs as a supportive tool until clear regulatory mechanisms are established.”
Additional Requirements:
Comment:
Effect sizes and confidence intervals should be reported in all tables.
Response: Thank you for this comment. As discussed in a response to an earlier comment, we have now added eta-squared values and CI figures to all relevant tables.
Comment:
The normality assumption of Kruskal-Wallis should be clearly justified.
Response: Thank you for this comment. We have now clarified in the statistical analysis section that Kruskal-Wallis tests are a non-parametric equivalent to ANOVA and would be used if assumptions of normality were violated (via Levene’s and Shapiro Wilk tests:
“Levene’s and Shapiro Wilk tests were conducted to determine normality of distribution and variance, and thus whether a parametric analysis of variance (ANOVA) or Kruskal-Wallis test (non-parametric equivalent) would be used to compare question and criterion-based differences. “
Comment:
Ensure that Dunn test p-values are adjusted for multiple comparisons.
Response: Thank you for this comment. All Dunn test results (Tables 2,3,4,5 and 7) report p values adjusted for multiple comparisons.
Comment:
Abstract needs clear statistical summary.
Response: Thank you for this comment. We believe that providing data on CI, effect size and p-values on all 10 questions would be unusual for an abstract, and moreover, doing so would render it too long. However, we have now specified the value we consider statistically significant ‘(p < 0.05)’ to the sentence, “Investigators found that neither ChatGPT-4o nor Chat GPT-4o1 preview were statistically outperformed (p < 0.05) by human dietitians on any of the study’s 10 questions.”
Comment:
Results should integrate figures/tables for clarity.
Response: Thank you for this comment. The study contains 7 tables and 2 figures, including a figure that we replaced in response to one of your previous comments. If you would like us to include any more, please specify the data you would like us to capture in these tables/figures.
Comment:
Discussion should emphasize clinical impact and AI ethics.
Response: Thank you for this comment. We have now added these 3 sentences to the end of the penultimate paragraph of the discussion section, which address clinical impact and AI ethics:
“It must be stressed, however, that this study does not provide evidence that Chat GPT-4o models can replace dietitians in real-world DWLSs. Ethical considerations concerning patient privacy, potential LLM algorithmic bias and general safety necessitate continued human oversight in all real-world weight loss interventions. As Aydin et al (2025) assert, LLMs offer multiple benefits to healthcare interventions, but clear regulatory boundaries are needed to ensure they “serve as supportive rather than standalone resources”[37].”
Comment:
The potential biases in AI responses should be explored.
Response: Thank you for this comment. We now mentioned potential AI biases at the end of the penultimate paragraph of the discussion section:
“Ethical considerations concerning patient privacy, potential LLM algorithmic bias and general safety necessitate continued human oversight in all real-world weight loss interventions.”
Comment:
Clearly state whether GPT-4o was tested for misinformation detection.
Response: Thank you for this comment. We do not believe this statement is necessary given that 4 assessors reviewed and scored every LLM response. However, if you feel strongly about it, we are more than happy to clarify that ‘misinformation detection was not used” in the discussion section.
Reviewer 3 Report
Comments and Suggestions for Authors
The manuscript compares ChatGPT versions 4o and 4o1 preview with human dietitian responses to a set of questions from medicated patients. The study included questions and responses from ChatGPT and dietitians were forwarded to four independent dietitians for assessment.
Please consider making the following improvements to the current version of the manuscript:
1) The second section "Materials and Methods" can start on page 3 instead of starting on page 4 and leaving almost the whole page 3 blank.
2) Please provide more details related to the detailed prompt described in section 2: "Investigators entered a detailed prompt into ChatGPT to describe the LLM’s role as a dietitian...".
3) The paragraph that starts with "Research manuscript reporting large dataset..." is not referenced, but contains the text found in other external sources (please check the attached Turnitin report).
4) Please reference the tests mentioned in section 2.2. and compare them. Explain why you decided to use the mentioned tests.
5) Please reference the tests mentioned in section 3.
6) LLMs sometimes respond differently to the same questions in different scenarios. Please elaborate on this as well.

Author Response
Comment:
The second section "Materials and Methods" can start on page 3 instead of starting on page 4 and leaving almost the whole page 3 blank.
Response: Thank you for noticing this. We have now reduced the space between the introduction and methods sections.
Comment: Please provide more details related to the detailed prompt described in section 2: "Investigators entered a detailed prompt into ChatGPT to describe the LLM’s role as a dietitian...".
Response: Thank you for this comment. The exact prompt is provided in Appendix A. We believe that the summary provided in the referred sentence (“describes the LLM’s role as a dietitian, the characteristics of a mock patient, and the response guidelines”) captures the essence of that prompt. However, if you feel strongly that further detail should be added to the sentence (or additional sentences), we are more than happy to include it.
Comment: The paragraph that starts with "Research manuscript reporting large dataset..." is not referenced, but contains the text found in other external sources (please check the attached Turnitin report).
Response: Thank you very much for noticing this. This was indeed text from the manuscript template that we had somehow failed to delete when inserting our study. The text has now been removed.
Comment: Please reference the tests mentioned in section 2.2. and compare them. Explain why you decided to use the mentioned tests.
Response: Thank you for this important point. We have now added references alongside the validation and correlation tests, along with an explanation that the KW test represents the non-parametric equivalent of the ANOVA:
“Median assessor scores were tabulated for all ten questions across the four respondents (two human dietitians and two LLMs). Levene’s and Shapiro Wilk tests were conducted to determine normality of distribution and variance [34], and thus whether a parametric analysis of variance (ANOVA) or Kruskal-Wallis test (non-parametric equivalent) would be used to compare question and criterion-based differences [35].”
In the 1st paragraph of the results section we clarify that a KW test was selected because an assumption of normality was violated (demonstrated through SW test):
“Scores from all four assessors across all four criteria were grouped for each question, and Levene’s and Shapiro Wilk tests were conducted [34]. The latter tests revealed that homogeneity of variance was violated, and thus a Kruskal-Wallis analysis was determined as the appropriate test for the study’s endpoints [35].”
Comment: Please reference the tests mentioned in section 3.
Response: Thank you for this suggestion. We have now added 2 citations to the first paragraph, where all tests were mentioned:
“Scores from all four assessors across all four criteria were grouped for each question, and Levene’s and Shapiro Wilk tests were conducted [34]. The latter tests revealed that homogeneity of variance was violated, and thus a Kruskal-Wallis analysis was determined as the appropriate test for the study’s endpoints [35].”
Comment: LLMs sometimes respond differently to the same questions in different scenarios. Please elaborate on this as well.
Response: Thank you for highlighting the vagueness of this statement. We have revised it to the following:
‘Studies have also noted that previous ChatGPT models have provided inconsistent responses to the same prompt over time, creating potential confusion, and that some of the sources they referenced were fake [27].”
Round 2
Reviewer 1 Report
Comments and Suggestions for Authors
The authors did not fully address the issues requested in the previous revision.
- It was recommended that a related studies section be added, but the authors passed this revision with the 2 paragraphs they added to the introduction section. One of these 2 paragraphs already includes the contribution of the study. In this revision, it is again suggested to add a related studies section to the study.
Author Response
Comment: The authors did not fully address the issues requested in the previous revision.
- It was recommended that a related studies section be added, but the authors passed this revision with the 2 paragraphs they added to the introduction section. One of these 2 paragraphs already includes the contribution of the study. In this revision, it is again suggested to add a related studies section to the study.
Response: Thank you for this suggestion and sincere apologies for misinterpreting it and thus failing to implement it to your satisfaction in the previous revision. We have now added a ‘related studies’ section to the introduction and expanded on our review of the literature on ChatGPT use in dietetics by adding 20 lines of relevant text (and multiple references).
The added text reads as follows:
“A 2025 systematic review of ChatGPT’s reliability in providing dietary recommendations found that fifteen studies had hitherto focused on the LLM’s performance in isolation, four had provided descriptive insights and five had compared the LLM with human dietitians [20]. Two of the comparative studies evaluated the accuracy with which ChatGPT models could adhere to established dietary guidelines, including the US Department of Agriculture’s dietary reference intake [21] and the Mayo Clinic Renal Diet Handbook for chronic kidney disease patients [22]. The former found that the LLM had difficulty catering to vegans [21], and the latter concluded that while ChatGPT-4 had outperformed its predecessor, it still made errors identifying potassium and phosphorous content in various foods [22]. Another study found that ChatGPT-4 accurately calculated protein content in 60.4% of foods item selected from a United Nations report [23]. The remaining two studies focused of the systematic review [20] compared ChatGPT and human responses to a list of obesity patient questions [13,19]. The first of these qualitatively analyzed responses to ten distinct prompts and concluded that the frequency with which ChatGPT 3.5 provided misleading information outweighed its access benefits [13]. The latter, a 2023 study by Kirk et al, quantitatively compared ChatGPT 3.5 and human dietitian responses to eight questions using a three-item metric (scientific correctness, comprehensibility, and actionability). The study reported that the LLM outperformed human dietitians across five of the study's eight questions and achieved comparable scores for the remaining three questions [19].”
Reviewer 2 Report
Comments and Suggestions for Authors
I thank the authors for revising the manuscript. The current version may be considered for the publication.
Author Response
Thank you kindly for your review.
Reviewer 3 Report
Comments and Suggestions for Authors
Dear Authors, thank you for accepting all suggestions for improvements. I think the manuscript can proceed to the following stages of the publishing process.
Author Response
Thank you kindly for your review.